# Health-promoting text messages to patients with hypertension—A randomized controlled trial in Swedish primary healthcare

Beata Borgström Bolmsjö[1,2]*, Jenny Bredfelt[1], Susanna Calling[1,2], Hanna Glock[1,2], Veronica Milos Nymberg[1,2], Kristina Bengtsson Boström[3], Ulf Jakobsson[1,2], Peter Nymberg[1,2], Jon Pallon[1,4], Mattias Rööst[1,4], Moa Wolff[1,2]

1 Center for Primary Health Care Research, Department of Clinical Sciences in Malmö, Lund University, Malmö, Sweden, 2 University Clinic Primary Care Skåne, Region Skåne, Sweden, 3 School of Public Health and Community Medicine, Institute of Medicine, Sahlgrenska Academy, University of Gothenburg, Gothenburg, Sweden, 4 Regional Department of Competence in Family Medicine and Primary Health Care, Kronoberg, Sweden

* beata.borgstrom-bolmsjo@med.lu.se

**Data Availability Statement:** Data cannot be shared publicly. Because of the regulations provided following the ethics application, it is not

## Abstract

Due to the high prevalence and great cardiovascular risks of hypertension, we need effective and evidence-based treatment strategies. Health-promoting one-way text messages could be a beneficial complement to antihypertensive drugs. However, this has yet to be proven in a primary healthcare setting. The purpose of this study was to investigate if health-promoting text messages could improve patients' blood pressure in primary care. The PUSHME (Primary care Usage of Health promoting Messages) randomized controlled trial included 401 patients from 10 primary health care centers in southern Sweden. Patients in the intervention group received four text messages weekly for six months along with treatment as usual. The PUSHME study was registered on clinicaltrials.gov (NCT04407962). Patients in both the control group and the intervention group lowered their blood pressure during the study, but there was no significant difference in change between the groups. However, subgroup analyses showed that there was a significantly larger reduction in diastolic blood pressure favoring the intervention for patients with poor self-rated health: -4.5 mmHg vs -1.4 mmHg (p = 0.019), and patients with a sedentary lifestyle: -5.2 mmHg vs -2.4 mmHg (p = 0.034). Our findings indicate that text messages with lifestyle advice to a general hypertensive population do not have any significant effect on blood pressure. However, it could be an effective complement to conventional antihypertensive drug treatment for specific patient groups.

## Introduction

Treatment of hypertension includes two well established strategies to lower blood pressure (BP): lifestyle interventions and drug treatment [1]. The World Health Organization (WHO) states that cost-effective blood pressure lowering lifestyle changes include reducing overweight,

possible to make the data public. The included patients have been informed that the results will only be published at group level and cannot be traced back to individuals. Providing data freely available to other researchers would make it possible to trace individual data and would breach compliance with the protocol approved by the Swedish Ethical Review Authority. E-mail address to Institutional point of contact who can field data inquiries from fellow researchers: registrator@etikprovning.se.

**Funding:** This study was funded by the Swedish Heart-Lung Foundation, the Swedish Southern Health Care Region and by Swedish governmental funding of clinical research (ALF) awarded to Susanna Calling. The funders had no role in study design, data collection and analysis, decision to publish, or preparation of the manuscript.

**Competing interests:** The authors have declared that no competing interests exist.

alcohol consumption and sodium intake and increasing physical activity and intake of fruits and vegetables [2]. Antihypertensive medication generally lowers the systolic blood pressure (SBP) by about 9 mmHg and the diastolic blood pressure (DBP) by about 5 mmHg [3]. However, despite effective medications, poor adherence to antihypertensive treatment remains a global problem [4]. About half of the patients prescribed an antihypertensive drug stopped taking it within one year [5].

Recommended strategies for improving blood pressure control include providing information and motivation for adopting lifestyle changes and adherence to prescribed medical treatments. Although there are clear guidelines regarding the importance of healthy lifestyle counseling, research shows that only 18% of Swedish physicians in primary healthcare (PHC) use these in a clinical context [6]. One explanation for this might be a strained PHC where lifestyle advice is expected to be delivered on top of other tasks. A way to solve this would be for lifestyle interventions to be delivered with less focus on medical staff involvement, and more on patient empowerment and engagement. An example of this could be text messages with lifestyle advice. Sending text messages is inexpensive, particularly when compared to the cost of in-person interventions or more sophisticated digital health tools. More advanced digital tools that provide interactivity and personalized care requires development, maintenance, and the need for patient education on how to use these technologies, limiting their usefulness in certain populations. With text messages, health care providers can reach many patients without significant additional infrastructure. Text message interventions are highly scalable, making them an attractive option for interventions for high prevalent conditions such as hypertension and cardiovascular disease [7]. Text messages can easily be automated, though with the limitation of being less personalized, allowing for the consistent delivery of reminders, lifestyle advice, and medication adherence prompts without the need for continuous healthcare professional involvement [8]. If one-way text messages can demonstrate significant improvements in blood pressure control, they could serve as a time-efficient and valuable supplement to standard care. Such an approach uses the patients' inherent motivation to comprehend and manage their hypertension, while avoiding additional strain on healthcare professionals.

Existing research on the integration of telehealth and text messages into lifestyle modification has yielded inconclusive findings [9]. To further explore this research area in a primary care setting, the present study was conducted at Swedish primary health care centers (PHCCs) examining a straightforward one-way short message service (SMS) intervention, seamlessly applicable in routine primary care.

The primary objective of this study was thus to examine how regularly sent text messages with standardized lifestyle advice and disease information affect blood pressure in a primary healthcare setting. A secondary aim was to examine the participants' experiences of receiving text messages with lifestyle advice.

## Materials and methods

This study, with the acronym PUSHME (Primary Care Usage of Health promoting Messages), was conducted as a randomized controlled multi-center study. A total of 401 patients were included in the study. The patients were recruited from 10 primary health care centers located in four different regions in the south of Sweden. The patients were included between September 1st 2020 and June 15th 2023.

### Inclusion

A list with all patients aged 40–85 years with a diagnosis of hypertension in the patient register was created for each PHCC. A random selection of patients from the list were invited to

participate in the study by postal mail. After 1–2 weeks they were contacted via phone by a specially trained research nurse that provided additional information about the study and could answer questions. If the patient agreed to participate, a baseline visit was scheduled at the patient's primary healthcare center. Written informed consent from each patient was collected at the baseline visit.

Inclusion criteria were patients with hypertension (defined by ICD-10, diagnose code I10.9), aged 40–85 years and owning a smartphone. Exclusion criteria were BP at baseline >180/110 mmHg or systolic blood pressure <120 mmHg, serious illness with short life expectancy (< 1 year) and predicted inability to comply with the study protocol e.g., language difficulties, cognitive impairment.

The primary outcome was change in blood pressure from baseline to follow-up. Power analysis based on the results from the pilot study [10], indicated that a sample size of 186 patients in each arm, was required. The calculation was based on an assumed statistical power of 80%, a two-sided test, using a significant level of 5% with a difference of 4 mm Hg between the groups a standard deviation of 13 mm Hg and a dropout rate of 10%.

## Randomization

After the completion of baseline assessments and questionnaires, the study nurse emailed information about the included patient to a researcher not connected to the patient and at a different site. Randomization was then performed by the researcher according to a computer-generated predefined block randomization for each study site. After randomization, information about which group the patient was included in was sent to the patient by postal mail. The study nurse and the primary care physicians were blinded to the patient's group allocation.

## Intervention

The patients in the intervention group received four weekly text messages with lifestyle advice for six months. Their regular anti-hypertensive treatment went on as usual. Every week, the participant received one text message from each of the following groups: A. physical activity, B. tobacco use, C. dietary habits and D. cardiovascular health in general, including alcohol use. Non-smokers received one extra text message, randomly selected from group A, C or D instead of the tobacco use text message. The text messages were developed by the authors based on the Swedish national guidelines for lifestyle habits [11, 12] and edited by experts on lifestyle habits in the regional healthcare administration Examples of the text messages are shown in Table 1. Messages were sent at random times between 9 AM and 7 PM. The control group received usual care according to the PHCC's regular practice. The text messages were refined after feedback from the participants in the pilot study and were personalized so that

**Table 1. Examples of test messages in the study, starting with the patient's first name.**

| | |
|---|---|
| *"Dear John! All adults are recommended to stay physically active, like brisk walks, for a total of at least 150 minutes per week. That's roughly half an hour a day for five days a week. Best Regards SMS Study Team"* | *"Dear John! Remember, sauces, broths, and flavor enhancers like ketchup, soy sauce, fish sauce, and stocks often have high salt content. See if you can find low-sodium alternatives or use different spices instead. Best Regards SMS Study Team"* |
| *"Dear John! High blood pressure accelerates the atherosclerosis of arteries, making it difficult for oxygenated blood to reach organs like the heart and brain. Atherosclerotic vessels can also release small blood clots that completely block arteries, leading to heart attacks or strokes. Blood pressure treatment effectively counters these risks. Best Regards, SMS Study Team"* | *"Dear John! Did you know that there are more than 60 different alcohol-related diseases? Alcohol can worsen conditions such as high blood pressure, overweight, and mood even with low consumption. Reducing alcohol intake usually leads to significant improvements relatively quickly. Best Regards, SMS Study Team"* |

each message started with the patient's first name and the non-smokers did not receive any messages about smoking.

## Measurements

Data were collected at the patients' PHCC by a study nurse at the baseline visit and six months later. Electronic BP monitors that were procured for healthcare centers in southern Sweden (Boso Medicus) were used to measure BP [13] on the right arm in a seated position after 5–10 minutes of rest. The following measurements were also included: heart rate, height, body weight, and waist circumference (for details on the measurements see S1 Data). BMI was calculated and grouped into normal weight ($\leq$24.9 kg/m$^2$), overweight (25.0–29.9 kg/m$^2$) and obese ($\geq$30.0 kg/m$^2$) [2].

Blood samples were collected at baseline visit and at the six-month follow-up for analysis of HbA1c, total cholesterol, LDL-cholesterol, and High-Density Lipoprotein-cholesterol (HDL-cholesterol) (details in S1 Data). However, if blood samples were already recorded within three months prior to the baseline visit at the PHCC, these were used as the baseline tests. Non-HDL-cholesterol was calculated as total cholesterol minus HDL-cholesterol.

At baseline and six-month follow-up, the patients also completed a short questionnaire assessing medications, medical history, family history of high BP, years since hypertension diagnosis, self-rated health (SRH), and physical activity level. Queries with three answer options (yes, no, don't know) were during analysis dichotomized into new variables with two answers: "Yes" (= yes) or "No", "I don't know" (= no and don't know). Level of education was dichotomized into "none or elementary school" or "upper secondary or higher education". The number of years since a patient's hypertension diagnosis were categorized into "Newly diagnosed" and ">5 years with hypertension diagnosis". SRH was measured with the question "How would you rate your general health?" with five response options on a Likert scale: very good, good, fair, poor, or very poor. For analysis, the response options "very good, good" were recoded into "good SRH" and the remaining response options into "poor SRH".

Physical activity was measured in minutes per week and divided by two questions into high- intensity exercise (e.g., jogging or aerobics) or low intensity exercise (e.g., walking or tranquil bicycling). Response options for high-intensity exercise ranged from 0 to >120 minutes per week and for low-intensity exercise from 0 to >300 minutes per week. These activities were then converted into activity minutes using the formula: activity minutes = 2*high-intensity exercise + low-intensity exercise activity [14]. Patients categorized as "physically active" reached the Public Health Agency of Sweden's recommended level of activity of 150 activity minutes per week whilst patients categorized as "sedentary" were below 150 minutes of activity [15].

## Medications

Current medications and potential alterations in drug treatment during the study were self-reported by the patients. The total number of drugs and the number of antihypertensive drugs per patient were calculated and usage of lipid-lowering drugs was noted. The antihypertensive drugs were categorized into eight different groups according to their mechanism of action. The patients with combination antihypertensive drugs, e.g., one pill containing both an angiotensin II receptor blocker (ARB) and a thiazide diuretic, were marked as having two antihypertensives. Changes in antihypertensive medications at follow-up were categorized into Altered medications or not. Altered medications included; "increased dose", "lowered dose", "newly prescribed", "discontinued" or "changed medication".

## Questionnaire about patients' experience of the study

After follow-up, the patients in the intervention group were asked to fill out an anonymous questionnaire about their study experiences. The questions included different statements regarding the text messages and how these affected the participants, their lifestyle habits and if the messages provided them with new knowledge. The participants were given four answer options for each statement; "fully agree", "strongly agree", "agree to a low degree", or "do not agree at all".

## Statistical analysis

Data were analyzed using IBM SPSS Statistics 29 (IBM Corp., Released 2023, Armonk, NY, USA). Calculations were conducted on the Full Analysis Set (FAS) data where all participants who were randomized into intervention or control and attended the follow up visit were accounted for. Differences between groups were calculated using Student's t-test for continuous variables, Mann-Whitney U-test for nonparametric variables and Chi-Square test for categorical variables. Change in blood pressure from baseline to follow-up was referred to as delta systolic BP (SBP) and delta diastolic BP (DBP) and was calculated as SBP (or DBP) at follow-up minus SBP (or DBP) at baseline. Patients with delta values for SBP or DBP larger than +/-3SD were considered outliers and excluded from analyses of BP change. A secondary intention-to-treat analysis with imputation of missing variables for patients lost-to-follow up, was also performed with the mean of the observed values for follow-up blood pressure for each group respectively. Subgroup analysis of delta BP was made for pre-specified subgroups that were expected to respond better to the intervention.

## Ethical considerations

The study was approved by the Swedish Ethics Review Authority (approval no 2019–06361 and no 2021–04819) and a regional consent for obtaining patient data (KVB 128–20). The PUSHME study was registered on clinicaltrials.gov (NCT03442257). The patients provided written consent at the baseline visit. All analyses were performed on pseudonymized data. Monitoring of the study was made at all study sites by an external expert in Good Clinical Practice, to ensure adherence to the study protocol. The study was conducted following the CONSORT (Consolidated Standard of Reporting Trials) guidelines [16].

# Results

After receiving written information about the study by postal mail, 1162 patients were contacted by phone. Of these, 502 patients were willing to participate in the study and were invited to a baseline visit (Fig 1). At randomization, 193 patients were allocated to the intervention group and 208 patients to the control group. At the six-month follow-up, there were 188 patients in the intervention group and 185 in the control group (Fig 1). No significant adverse effects or unintended consequences were observed in any of the groups.

## Baseline characteristics

The baseline characteristics for the control and the intervention group, respectively, are presented in Table 2 and did not differ significantly in any aspect. The mean age was 68.6 years. In total, 47.6% of the patients were women, and the mean BMI was 28.6 kg/m2. There were no significant differences between groups regarding mean SBP and mean DBP at baseline. Almost 66% of the participants stated a family history of high BP. Around 30% of patients had a sedentary lifestyle with activity minutes < 150 minutes weekly.

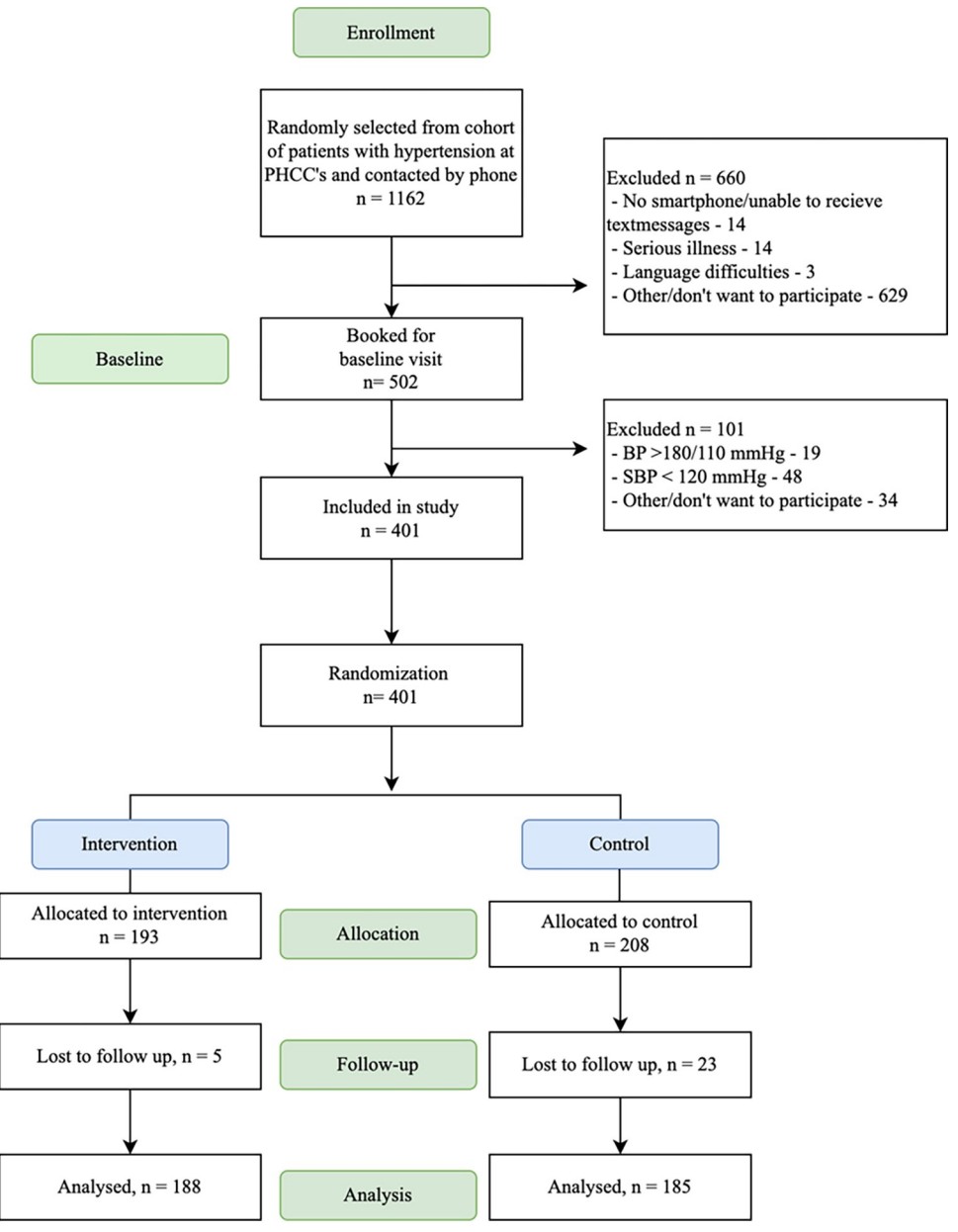

**Fig 1. Flowchart of inclusion of patients with hypertension in the PUSHME study.**

## Medications

The mean number of medications at baseline was 4.3 and the mean number of antihypertensive medications was 1.9, with no significant differences between the groups (Table 3). The use of different classes of antihypertensives did not differ significantly between the groups except for beta-blockers (39.4% in the control group compared to 27.5% in the intervention group, p = 0.011). The number of patients who altered their antihypertensive treatment from baseline to follow-up did not significantly differ between the groups (n = 27 in the control group and n = 24 in the intervention group). In both groups, the most common alteration of antihypertensive medication was increased dose. The use of lipid-lowering medications at baseline was 45.1% and did not differ between the groups.

**Table 2. Baseline characteristics.**

| | Total | Control | Intervention |
|---|---|---|---|
| | n = 401 | n = 208 | n = 193 |
| Female sex, n (%) | 191 (47.6) | 103 (49.5) | 88 (45.6) |
| Age, years | 68.6 ± 9.4 | 69.0 ± 9.8 | 68.2 ± 8.9 |
| BMI, kg/m$^2$ | 28.6 ± 5.0[1] | 28.7 ± 5.1 | 28.4 ± 4.8[1] |
| Waist circumference, cm: | | | |
| • Women | 97.2 ± 13.3[1] | 97.6 ± 13.4 | 96.7 ± 12.6[1] |
| • Menv | 106.4 ± 11.8 | 106.8 ± 11.5 | 106.1 ± 12.2 |
| SBP, mmHg | 140.5 ± 13.0 | 140.5 ± 13.1 | 140.4 ± 13.0 |
| DBP, mmHg | 84.1 ± 10.0 | 84.6 ± 10.6 | 83.5 ± 9.4 |
| HR, beats/min | 69.2 ± 11.5 | 69.5 ± 11.5 | 69.0 ± 11.5 |
| Altered BP-medication? Yes, n (%) | 51 (13.7) | 27 (14.6) | 24 (12.8) |
| HbA1c, mmol/mol | 39.6 ± 6.7 | 39.3 ± 6.6 | 40.1 ± 6.8 |
| Non-HDL cholesterol, mmol/L | 3.4 ± 1.1[1] | 3.5 ± 1.1[1] | 3.4 ± 1.2 |
| Family history of high BP, n (%) | 264 (65.8) | 136 (65.4) | 128 (66.3) |
| Sedentary lifestyle, n (%) | 128 (31.9) | 66 (31.7) | 62 (32.1) |
| Previous CVD, n (%) | 58 (14.5) | 30 (14.4) | 28 (14.5) |
| Upper secondary / higher education, n(%) | 292 (72.8) | 153 (73.6) | 139 (72.0) |
| >5 years hypertension diagnosis, n (%) | 272 (67.8) | 143 (68.8) | 129 (66.8) |
| Good SRH, n (%) | 280 (69.8) | 144 (69.2) | 136 (70.5) |

Normally distributed variables are displayed as mean ± SD. BMI: body mass index, SBP: systolic blood pressure, DBP: diastolic blood pressure, HR: heart rate, Sedentary lifestyle: <150 activity minutes/week, Previous CVD: previous cardiovascular disease (previous stroke, previous myocardial infarction, angina pectoris, aortic aneurysm, previous operation CABG or angioplasty), SRH: self-rated health, [1]One patient's data missing

### Change in BP

There was no difference in BP change between control and intervention in the total study population (SBP control vs intervention -3.8 mmHg vs -3.9 mmHg (p = 0.98) and for DBP -2.1 mmHg vs -3.2 mmHg (p = 0.17) (Table 4). There was still no difference between groups after imputing the lost-to follow up values (SBP control vs intervention -4.3 mmHg vs -3.7 mmHg (p = 0.63) and for DBP -2.2 mmHg vs -3.3 mmHg (p = 0.17)).

### Subgroup analysis on change in BP

Subgroup analyses on change in BP were made for specific subgroups (male sex, participants classed as sedentary and participants with poor SRH at baseline) and are presented in Table 4. For patients who reported a sedentary lifestyle at baseline, the intervention group had a larger reduction in DBP with -5.2 mmHg vs. -2.4 mmHg in the control group (p = 0.03), and a difference in mean change between control and intervention groups of -2.8 (95% CI: -5.4 to -0.2). For individuals with poor SRH at baseline, the intervention group also had a greater reduction of DBP than the control group (-4.5 vs -1.4 mmHg (p = 0.02)) (Table 4 and Fig 2).

### Questionnaire about patients' experience of the study

A total of 184 patients in the intervention group received the questionnaire, and 54 of them responded, sharing their experiences of participating in the PUSHME study. The questionnaire was exclusively given to patients in the intervention group. More than 90% of the participants reported that they read all text messages, as illustrated in Fig 3. Most of the patients

**Table 3. Medications.**

| | Total | Control | Intervention |
|---|---|---|---|
| **Baseline** | n = 401 | n = 208 | n = 193 |
| Total number of medications | 4.3 ± 2.6 | 4.4 ± 2.6 | 4.3 ± 2.5 |
| Total number of BP-medications | 1.9 ± 0.9 | 1.9 ± 0.9 | 1.8 ± 0.9 |
| Number of BP-medications: | n (%) | | |
| • 0 | 11 (2.7) | 5 (2.4) | 6 (3.1) |
| • 1 | 145 (36.1) | 72 (34.6) | 73 (37.8) |
| • 2 | 149 (37.1) | 73 (35.1) | 76 (39.4) |
| • >2 | 96 (23.9) | 58 (27.9) | 38 (19.7) |
| Class of BP medications: | n (%) | | |
| • ARB | 216 (53.8) | 112 (53.8) | 104 (53.9) |
| • ACE inhibitor | 92 (22.9) | 50 (24.0) | 42 (21.8) |
| • Calcium channel blocker | 166 (41.4) | 82 (39.4) | 84 (43.5) |
| • Beta blocker | 135 (33.7) | 82 (39.4) | 53 (27.5) |
| • Thiazide diuretic | 88 (21.9) | 47 (11.7) | 41 (21.2) |
| • Loop diuretic | 17 (4.2) | 8 (3.8) | 9 (4.7) |
| • Alpha blocker | 17 (4.2) | 6 (2.9) | 11 (5.7) |
| • Aldosterone antagonist | 13 (3.2) | 10 (4.8) | 3 (1.6) |
| Use of lipid-lowering med., n (%) | 181 (45.1) | 94 (45.2) | 89 (46.1) |
| **Follow-up** | n = 372 | n = 185 | n = 187[1] |
| Altered BP-medication? Yes, n (%) | 51 (13.7) | 27 (14.6) | 24 (12.8) |

Normally distributed variables displayed as mean ± SD. BP; blood pressure, ARB; angiotensin receptor blocker, ACE; angiotensin-converting enzyme inhibitor.

[1]One patient's data missing

agreed that the text messages served as a helpful reminder of good lifestyle habits. They also found that the intervention provided increased knowledge about good lifestyle habits. Regarding the effect on different lifestyle changes, the patients agreed most on that the text messages affected their dietary habits, and agreed least on that the messages had any impact on their alcohol habits.

**Table 4. Mean change in blood pressure.**

| | SBP | | | | DBP | | | |
|---|---|---|---|---|---|---|---|---|
| | Control | Interven-tion | p | [a]Difference in mean change (95% CI) | Control | Interven-tion | p | [a]Difference in mean change (95% CI) |
| **All** | n = 181 -3.8±13.7 | n = 185 -3.9±2.9 | 0.98 | -0.03 (-2.8 to 2.7) | n = 181 -2.1±7.6 | n = 185 -3.2±6.9 | 0.17 | -1.1 (-2.6 to 0.5) |
| **Sex**, men | n = 93 -4.2±13.3 | n = 102 -3.5±13.3 | 0.67 | 0.8 (-2.9 to 4.5) | n = 93 -2.6±7.7 | n = 102 -3.3±7.2 | 0.49 | -0.7 (-2.8 to 1.4) |
| **Seden-tary** | n = 57 -2.7±14.6 | n = 58 -3.6±14.1 | 0.74 | -0.9 (-6.2 to 4.4) | n = 57 -2.4±7.7 | n = 58 -5.2±6.3 | **0.034** | -2.8 (-5.4 to -0.2) |
| **SRH poor** | n = 55 -2.2±14.5 | n = 53 -3.2±13.0 | 0.35 | -1.1 (-6.3 to 4.2) | n = 55 -1.4±7.6 | n = 53 -4.5±7.8 | **0.019** | -3.1 (-6.1 to -0.8) |

Individuals with ±3SD of the delta value (follow-up SBP/DBP–baseline SBP/DBP) are removed from the analysis, n = 6. Presented as mean ±SD (mmHg). SBP, systolic blood pressure; DBP, diastolic blood pressure; BMI, body mass index. Analyzed with independent T-test. Level of significance: **p <0.05**

[a]Difference between mean change in Intervention and mean change in Control groups

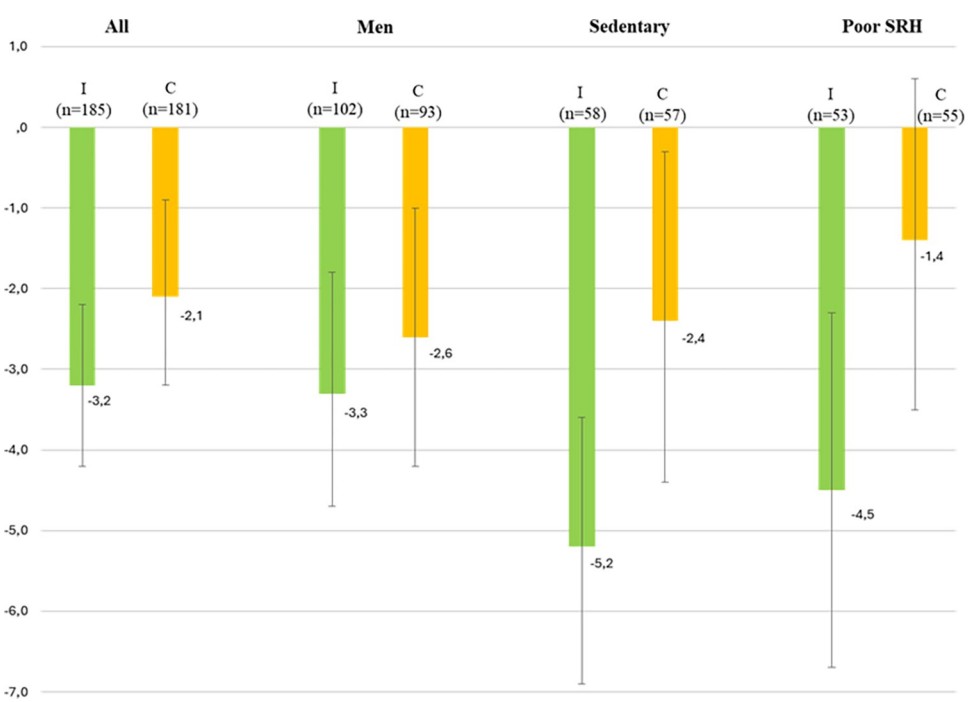

**Fig 2. Mean change from baseline to follow up in diastolic blood pressure (mmHg).** 95% CI, (I = Intervention group, C = Control group).

## Discussion

This study on text messages with lifestyle advice and cardiovascular information to Swedish primary health care patients with hypertension showed no statistically significant difference in blood pressure reduction between the intervention and the control group. However, statistically significant larger reductions in DBP were found in the specific subgroups; patients with poor SRH and patients with a sedentary lifestyle. Possible explanations for these positive effects

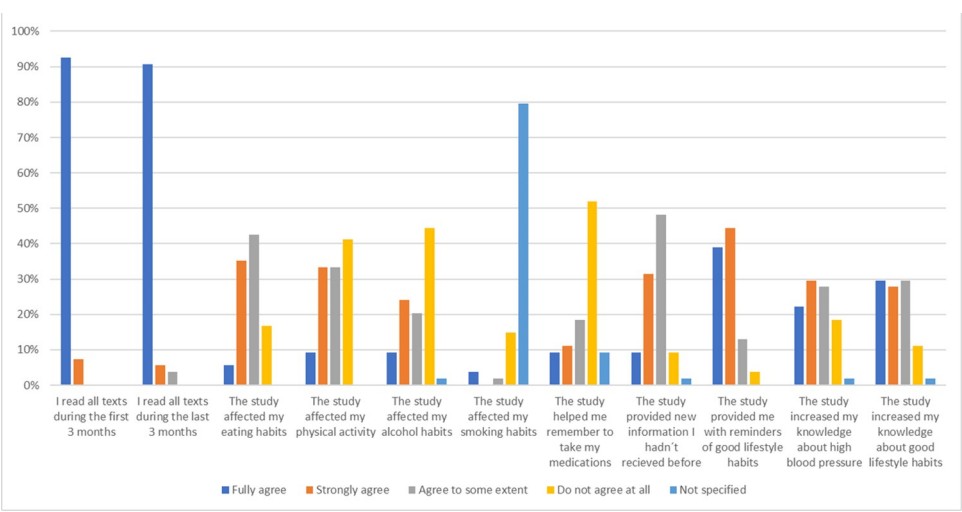

**Fig 3. Questionnaire about patients' experience of the study n = 54.**

in the subgroups include encouragement of improved lifestyle habits as well as increased knowledge of hypertension, cardiovascular disease, and the importance of treatment.

The use of telehealth to influence lifestyle behavior has been a recurrent topic of discussion and research in recent years. Meta-analyses have yielded contradictory results regarding the effects of digital interventions on blood pressure, possibly due to the considerable heterogeneity of included studies [17–19]. A beneficial effect can also be dependent on prior events such as in the TEXTME study where the participants in the study had proven coronary heart disease (for example prior myocardial infarction), and therefore may have been more motivated to lifestyle change [20]. Another plausible explanation for the differing effects may stem from the fact that various digital interventions prove effective for certain patient subgroups while proving less beneficial for others, as our study illustrates. Specifically, our study highlights significant reductions with confidence intervals indicating potentially meaningful clinical benefit in diastolic blood pressure (DBP) favoring the intervention, within specific subgroups such as sedentary patients and those reporting poor self-rated health (SRH). As a reduction of 5 mmHg in DBP is linked to a noteworthy decrease in all major cardiovascular events [21] our findings where the patients with a sedentary lifestyle reduced their DBP by 5.2 mmHg indicate that this intervention could be a meaningful contribution to reducing cardiovascular risk in certain subgroups. Noteworthy is also that the effect of the intervention in sedentary patients and in patients with poor SRH is comparable to the effect of one antihypertensive medication, which is important from a clinical perspective. Further, evidence suggests that persons with low SRH and those with a sedentary lifestyle are hard to engage in lifestyle changes [22]. As patients with a sedentary lifestyle and low SRH also have a higher cardiovascular risk [23, 24] our intervention might be able to contribute to decreased health inequity.

A possible reason for why significant differences were found in DBP but not SBP could be the larger standard deviations found in the measures of SBP as the effect sizes relative to the variability in the data then become less pronounced. SBP is known to have a larger variability than DBP [25]. Accurate blood pressure measurement has also been emphasized as being often overlooked in clinical practice, while it could have a considerable impact on management strategies [26]. In future studies one could consider using 24h BP measurements, as patients may have a larger variability in office BP due to stress or anxiety in the clinical setting, and more BP readings measured over a 24-hour period would give a more credible result. Although as both increased SBP and DBP have been showed to be independent risk factors of CVD, also lowering of only DPB should have good clinical relevance [27].

The results from the questionnaire indicate that although the patient has the information and understands the health benefits of good habits, it is hard to convert that information into action. Health professionals could promote lifestyle changes in many ways, but it will not matter if the patients are not motivated to make the changes [22]. The previously published PUSHME interview study regarding patient experiences concluded that timing in relation to diagnosis and tailoring the messages to patients' prior knowledge and habits could increase the effectiveness of the messages [28]. These results in combination with the results from our questionnaire provide important information on how to improve the intervention and indications on why some patients might have experienced difficulties when trying to translate knowledge into action.

This study was preceded by a published pilot study demonstrating good feasibility [10]. Utilizing insights from this study, the text messages were refined based on feedback from participating patients. Another strength of this study is that it was conducted in a primary care setting as a randomized clinical trial with 401 recruited patients. The number of the allocated patients (193 vs 208) was slightly imbalanced due to the use of specific allocation sequences at the different sites. This procedure may create imbalances despite proper randomization.

However, the randomization of patients was successful with no significant differences between intervention and control at baseline. Included PHCCs were also recruited from different regions of southern Sweden with the aim of including patients from urban and rural areas and from different socioeconomic groups.

The text messages used in this study were designed with inspiration from motivational interviewing—a technique proven to help patients find motivation for changing unhealthy habits, which also could be seen as a strength [23]. Furthermore, we left the responsibility to perform the changes to the patients. We provided participants with information about the positive effects of a good lifestyle and some tips on how to achieve this but provided no gym classes, meetings with dietitians or such to physically help them. This approach where patients are left with a big responsibility to change their own habits is a part of the patient empowerment theory where they gain power through acquiring knowledge. This has previously been proven to be one way of lowering SBP and DBP in patients with hypertension [24].

This study has some limitations. First, the number of patients lost-to-follow-up was imbalanced between the groups. In the control group 23 participants did not show up for the follow-up in contrast to only five in the intervention group. The reason for this might be that the participants in the control group were not as designated to measure blood pressure after six months since they hadn't received any lifestyle advice. This may have influenced the results as we might have lost the some of the participants who did not improve their lifestyle habits. However, scenarios with imputations of lost to follow-up values did not affect the outcome, suggesting that results were robust. Still, the bias introduced by differences in dropout rate is an important limitation of the study. Also, the Hawthorne effect [29], in which individuals regardless of group allocation modify their behavior or performance in response to being observed, could have contributed to the lack of large outcome differences between the groups. Stratifying the randomization by center alone could be seen as a limitation. We deemed center to be the most important variable to stratify by to control for center-specific effects, especially since the timing of inclusion differed between the centers. The patients could not be blinded to group allocation but were specifically told not to reveal the group belonging to their GP or nurse at the PHCC. There was no means of checking if this 'secondary' blinding of the GP was thorough. Additionally, the study design did not account for the possibility that participants in different groups might be family members or friends. This could have led to contamination of the control group if individuals in the intervention group shared their text messages.

Due to the data not meeting the normality assumptions necessary for ANCOVA and regression analyses, we opted to assess differences in mean changes using two-sided t-tests. This approach, however, precluded us from performing interaction tests to differentiate between chance effects in subgroups, representing a limitation of our study. Given the small number of subgroups, the risk of a Type I error was relatively low, and overcorrection could have resulted in the loss of valuable insights. Consequently, we have presented unadjusted p-values to offer a more balanced and informative perspective on the data. The confidence interval suggests a potential for a clinically beneficial effect, but the effect could also be minor or negligible. With this uncertainty, it is crucial to interpret these findings with caution and account for the degree of multiplicity when considering subgroup effects.

Unfortunately, the questionnaire data was not linked to the individual participant in the study, and therefore it was not possible to check how the response rate varied with outcome.

## Conclusion

This study does not prove any effect on BP for lifestyle advice through text messages in a general primary care population with hypertension. However, specific subgroup analysis suggests

that text messages could be a good complement to conventional hypertension treatment, with statistically significant larger mean reductions in DBP in certain subgroups. Targeted towards the right patients, this intervention could be beneficial and may extend beyond a reduction in blood pressure to also include a healthier lifestyle and the additional positive aspects associated with it.

Additional studies are needed to understand where the resources regarding lifestyle advice should be prioritized.

## Supporting information

**S1 Checklist. CONSORT 2010 checklist of information to include when reporting a randomised trial**\*.
(DOC)

**S1 Data.**
(PDF)

**S1 File. English translation of swedish summary.**
(DOCX)

**S2 File. Project plan_PUSHME.**
(DOC)

## Acknowledgments

We would like to acknowledge the study participants, the staff at the participating health care centers and 21st Century Mobile for delivering the text message service. We also want to thank Patrick O'Reilly for language editing.

## Author Contributions

**Conceptualization:** Beata Borgström Bolmsjö, Susanna Calling, Veronica Milos Nymberg, Kristina Bengtsson Boström, Moa Wolff.

**Data curation:** Beata Borgström Bolmsjö, Jenny Bredfelt.

**Formal analysis:** Beata Borgström Bolmsjö, Susanna Calling, Hanna Glock, Ulf Jakobsson.

**Funding acquisition:** Susanna Calling.

**Investigation:** Moa Wolff.

**Methodology:** Beata Borgström Bolmsjö, Susanna Calling, Hanna Glock, Veronica Milos Nymberg, Ulf Jakobsson, Moa Wolff.

**Project administration:** Beata Borgström Bolmsjö, Susanna Calling, Kristina Bengtsson Boström, Peter Nymberg, Jon Pallon, Mattias Rööst, Moa Wolff.

**Resources:** Susanna Calling.

**Supervision:** Beata Borgström Bolmsjö, Susanna Calling.

**Writing – original draft:** Beata Borgström Bolmsjö, Jenny Bredfelt, Susanna Calling.

**Writing – review & editing:** Beata Borgström Bolmsjö, Jenny Bredfelt, Susanna Calling, Hanna Glock, Veronica Milos Nymberg, Kristina Bengtsson Boström, Ulf Jakobsson, Peter Nymberg, Jon Pallon, Mattias Rööst, Moa Wolff.

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
