## [Decision Letter · Decision Letter 0]

12 Aug 2024

PONE-D-24-17656Health-promoting text messages to patients with hypertension - a randomized controlled trial in Swedish primary healthcarePLOS ONE

Dear Dr. Borgström Bolmsjö,

Thank you for submitting your manuscript to PLOS ONE. After careful consideration, we feel that it has merit but does not fully meet PLOS ONE’s publication criteria as it currently stands. Therefore, we invite you to submit a revised version of the manuscript that addresses the points raised during the review process.

We look forward to receiving your revised manuscript.

Kind regards,

Johanna Pruller, Ph.D.

Associate Editor

PLOS ONE

Journal Requirements:

"This study was funded by the Swedish Heart-Lung Foundation, the Swedish Southern Health Care Region and by Swedish governmental funding of clinical research (ALF) awarded to Susanna Calling."

3. In the online submission form, you indicated that [The data underlying the results presented in the study are available from the corresponding author on reasonable request]. 

Reviewers' comments:

Reviewer's Responses to Questions

**Comments to the Author**

1. Is the manuscript technically sound, and do the data support the conclusions?

Reviewer #1: Partly

Reviewer #2: Yes

2. Has the statistical analysis been performed appropriately and rigorously? 

Reviewer #1: Yes

Reviewer #2: Yes

3. Have the authors made all data underlying the findings in their manuscript fully available?

Reviewer #1: Yes

Reviewer #2: Yes

4. Is the manuscript presented in an intelligible fashion and written in standard English?

Reviewer #1: Yes

Reviewer #2: Yes

5. Review Comments to the Author

Reviewer #1: The authors here report on a randomised controlled trial of 401 individuals to receive or not one-way text messages to improve hypertension.

It is not impossible that two people in the same family or household could be hypertensive; how was this dealt with in the design of the study as these people would not escape contamination and would not be properly independent.

THe sample size is not quite right as it stands - without dropout a standardised difference would be 4/13, requiring 167 patients per arm; with 15% dropout we divide by 0.85 to get 197 per arm. Please check the calculations here - was there some rounding going on?

Please justify not using variables other than centre to stratify the randomisation - presumably time since diagnosis would be an important variable in terms of modification of lifestyle.

Drop out from the study is strongly related to treatment arm potentially introducing bias - please explore dropout (p=0.0009 on figures given line 207-210).

If randomisation has been performed correctly, then any differences at baseline must be down to chance and as such testing in table 2,3 (top) is not only unnecessary but can draw attention to a belief the randomisation is in some way problematic. Please remove.

In Table 4 please give absolute effect sizes and confidence intervals for differences between groups. Change in medication is an outcome and not a baseline covariate; all subgroup tests require tests for interaction to distinguish chance effects in subgroups. Please consider the degree of multiplicity in Table 4 before overemphasizing subgroups. To what extend does the data rule out a meaningful benefit based upon the confidence intervals?

How many people were approached with the questionnaire; was it all people in the intervention group? How does response rate vary by outcome?

Reviewer #2: Borgström Bolmsjö B et al. conducted a trial investigating the impact of health-promoting text messages on patients with hypertension. Their findings indicate that text messages containing lifestyle advice did not significantly affect blood pressure in the general hypertensive population. However, for specific subgroups—namely patients with poor self-rated health (SRH) and those with a sedentary lifestyle—the text messages served as an effective complement to conventional hypertension treatment. These subgroups experienced statistically significant larger mean reductions in diastolic blood pressure (DBP).

Below are some points that should be considered to improve the paper.

- In the Introduction, you mention: "However, despite effective medications, poor adherence to antihypertensive treatment remains a global problem (4)." I suggest also addressing the often-overlooked importance of accurate blood pressure measurement, as this has a significant impact on management strategies (cite PMID: 22902872).

- The randomized controlled trial design is appropriate for the research question. The inclusion criteria and the intervention protocol are clearly defined. It would be useful to include more information on how the text messages were created and adapted to individual patients.

- To enhance clarity, I recommend including graphical representations of blood pressure variations for both the intervention and control groups. Line graphs or bar charts depicting changes in systolic and diastolic blood pressure over time would be beneficial. Additionally, graphs showing diastolic blood pressure changes in subgroups, such as patients with poor self-rated health (SRH) and those with a sedentary lifestyle, would highlight the significant reductions observed.

- The manuscript would benefit from a more detailed discussion on the cost-effectiveness of text message interventions in hypertension management, particularly in comparison to other intervention strategies.

- Page 15: “First, the number of patients loss-to-follow-up were In the control group”. Please correct.

6. PLOS authors have the option to publish the peer review history of their article (what does this mean?). If published, this will include your full peer review and any attached files.

Reviewer #1: No

Reviewer #2: No

---

## [Author Response · Author response to Decision Letter 0]

12 Sep 2024

Dear Associate Editor, Johanna Pruller

Thank you for the opportunity to revise our manuscript titled “Health-promoting text messages to patients with hypertension - a randomized controlled trial in Swedish primary healthcare”. 

We appreciate the time and effort spent by You and the Reviewers on the manuscript and the valuable comments. 

Below, we have replied to the Journal Requirements and the Reviewers’ comments and taken actions point by point. All changes to the manuscript are marked as Track Changes in the document in the file labeled 'Revised Manuscript with Track Changes'.

We are looking forward to your response.

Yours sincerely,

Dr. Beata Borgström Bolmsjö, corresponding author

On behalf of all authors (who have seen and approved the final version of the manuscript)

Journal Requirements

Response: 

We believe that the manuscript and file naming now meet PLOS ONE’s style requirements. 

"This study was funded by the Swedish Heart-Lung Foundation, the Swedish Southern Health Care Region and by Swedish governmental funding of clinical research (ALF) awarded to Susanna Calling." Please state what role the funders took in the study. If the funders had no role, please state: "The funders had no role in study design, data collection and analysis, decision to publish, or preparation of the manuscript." If this statement is not correct you must amend it as needed. Please include this amended Role of Funder statement in your cover letter; we will change the online submission form on your behalf.

Response:

We have added this amended Role of Funder to the cover letter accordingly:

“This study was funded by the Swedish Heart-Lung Foundation, the Swedish Southern Health Care Region, and by Swedish governmental funding of clinical research (ALF) awarded to Susanna Calling. The funders had no role in study design, data collection and analysis, decision to publish, or preparation of the manuscript.”

3. In the online submission form, you indicated that [The data underlying the results presented in the study are available from the corresponding author on reasonable request]. 

Response: 

This is unfortunate but based on the regulations provided following the ethics application, it is not possible to make the data public. The included patients have been informed that the results will only be published at the group level and cannot be traced back to individuals. Providing data freely available to other researchers would make it possible to trace individual data and would breach compliance with the protocol approved by the Swedish Ethical Review Authority. Email: registrator@etikprovning.se

Reviewers’ comments

Reviewer #1: The authors here report on a randomised controlled trial of 401 individuals to receive or not one-way text messages to improve hypertension.

1.1. It is not impossible that two people in the same family or household could be hypertensive; how was this dealt with in the design of the study as these people would not escape contamination and would not be properly independent.

Response: 

Thank you for this point of view. The study design did not take this into concern, nor the possibility that two friends or other relatives were involved in the study, which also could create contamination. In small societies, it may have been the ‘talk of the town’. This is actually what we would like to create with the intervention in the future by increasing public awareness of lifestyle intervention in hypertension since it is such a common diagnosis. Still, this is a weakness of the study and may have influenced the participants in the control group in addition to the Hawthorne effect already mentioned in the manuscript. We have now added this to the Discussion, page 18 line 367-369.

“Additionally, the study design did not account for the possibility that participants in different groups might be family members or friends. This could have led to contamination of the control group if individuals in the intervention group shared their text messages.”

1.2. The sample size is not quite right as it stands - without dropout a standardised difference would be 4/13, requiring 167 patients per arm; with 15% dropout we divide by 0.85 to get 197 per arm. Please check the calculations here - was there some rounding going on?

Response:

Thank you for this careful observation. Yes, there has been a mistake in the figures. When conducting the pilot study, we discovered that the dropout rate was less than 5 %. Therefore, we chose to use a dropout rate in this full-scale study of 10%. This gives us 186 per arm. This is now corrected on page 5, line 109-110 and 113.

1.3. Please justify not using variables other than centre to stratify the randomisation - presumably time since diagnosis would be an important variable in terms of modification of lifestyle.

Response: 

Thank you for this relevant comment. The decision to stratify by center alone was driven by the need to control for center-specific effects, such as differences in patient demographics and healthcare practices, and to ensure that the number of patients in total would be evenly distributed between the groups. Since the sites were including participants on an ongoing basis during the COVID-19 pandemic the inclusion was conducted in different tempo at the different sites. By stratifying by center, the study ensured that any potential variability introduced by these factors was evenly distributed across groups.

Including additional variables like time since diagnosis in the stratification process would add complexity to the randomization scheme. The complexity of managing such a stratified randomization process we believed would outweigh the potential benefits.

We agree that time since diagnosis is an important variable for the intervention, but we believed that the randomization process should distribute participants with varying times since diagnosis relatively evenly between groups. This was confirmed and is shown in table 2. This has been added to the discussion page 18, line 359-364:

“That we stratified the randomization by center alone could be considered a limitation. We deemed center to be the most important variable to stratify by to control for center-specific effects, but also because the timing of inclusion differed between the centers. Including additional variables in the stratification process would have added a complexity to the randomization process that we consider would have outweighed the potential benefit.”

1.4. Drop out from the study is strongly related to treatment arm potentially introducing bias - please explore dropout (p=0.0009 on figures given line 207-210).

Response:

It is correct and very important to emphasize that the dropout differed largely between the groups. We agree that this could introduce bias to the results, and therefore this is the first and most important issue in the section about limitations in the Discussion. We have also made imputations, listed in the Results section, to try to adjust for the lost-to follow up. The results still seem robust as the scenarios with imputations of lost-to-follow-up values did not change the results. However, this bias cannot and should not be ignored. We have tried to clarify this further in the Discussion page 17 line 347-355. 

“First, the number of patients lost-to-follow-up was imbalanced between the groups. In the control group 23 participants did not show up for the follow-up in contrast to only five in the intervention group. The reason for this might be that the participants in the control group were not as designated to measure blood pressure after six months since they hadn’t received any lifestyle advice. This may have influenced the results as we might have lost some of the participants who did not improve their lifestyle habits. However, scenarios with imputations of lost to follow-up values did not affect the outcome, suggesting that results were robust. Imputing a worst/best case analysis could have benefited the intervention, however, that would only be speculations. Still, the bias introduced by differences in dropout rate is an important limitation of the study”

1.5. If randomisation has been performed correctly, then any differences at baseline must be down to chance and as such testing in table 2,3 (top) is not only unnecessary but can draw attention to a belief the randomisation is in some way problematic. Please remove.

Response: 

Thank you for this comment. We agree that it is redundant to show the p-values in table 2 and 3 and we have now removed that column from each of the tables. 

1.6. In Table 4 please give absolute effect sizes and confidence intervals for differences between groups. Change in medication is an outcome and not a baseline covariate; all subgroup tests require tests for interaction to distinguish chance effects in subgroups. Please consider the degree of multiplicity in Table 4 before overemphasizing subgroups. To what extend does the data rule out a meaningful benefit based upon the confidence intervals?

Response:

Thank you for these suggestions to strengthen the statistics of the data.

We have now added effect size and CI for differences between groups to the revised table 4. 

We agree that variable “change in medication”is an outcome rather than a baseline covariate. We have therefore removed this subgroup from table 4.

Regarding interaction: Unfortunately, our data did not meet the normality assumptions required for ANCOVA/regression analyses. Therefore, we analyzed differences in mean change using two-sided t-tests within different subgroups. This limitation prevented us from conducting interaction analysis on the subgroups, we have added this to the limitations in the Discussion page 17, line 370-373

“Due to the data not meeting the normality assumptions necessary for ANCOVA and regression analyses, we opted to assess differences in mean changes using two-sided t-tests. This approach, however, precluded us from performing interaction tests to differentiate between chance effects in subgroups, representing a limitation of our study.” 

Regarding multiplicity and meaningful benefit based upon the confidence intervals: Given the small number of subgroups, the risk of type I error is relatively low, and overcorrection could lead to the loss of meaningful insights. The benefit of adjusting p-values in this context is less clear, as the chance of a significant result due to random variation alone is already limited. Therefore, we have chosen not to adjust p-values for our three subgroups. We believe that presenting unadjusted p-values provides a more balanced and informative view of the data, while still recognizing the need for cautious interpretation. 

As a reduction of 5 mmHg in DBP is linked to a noteworthy decrease in all major cardiovascular events, and a meaningful and reasonable level of blood pressure change within a health intervention was set to 4 mmHg in the power calculation. Although the lower confidence intervals for the subgroups are close to zero, the level of meaningful benefits are within the confidence intervals and consistent effects with similar CIs across subgroups strengthen the overall interpretation. The confidence interval indicates that although there is a possibility of a practically significant effect, one must also consider that the effect could be less significant or even negligible. This uncertainty should be taken into account when interpreting and using the results with caution, especially as being subgroup analyses However, we have now added the risk of overemphasizing subgroups in the Discussion page 17, line 374-379. 

“Given the small number of subgroups, the risk of a Type I error was relatively low, and overcorrection could have resulted in the loss of valuable insights. Consequently, we have presented unadjusted p-values to offer a more balanced and informative perspective on the data. The confidence interval suggests a potential for a clinically beneficial effect, but the effect could also be minor or negligible. With this uncertainty, it is crucial to interpret these findings with caution and account for the degree of multiplicity when considering subgroup effects”

1.7. How many people were approached with the questionnaire; was it all people in the intervention group? How does response rate vary by outcome?

Response:

Thank you for important input on the part of the questionnaire. The questionnaire was sent out to 184 participants after the study and could not be traced to individual data. This is added to the Methods section, page 9 line 191.

 “The patients in the intervention group were after follow-up asked to fill in an anonymous questionnaire about their experiences of the study”

 and the Results page line 274. 

“A total of 184 patients in the intervention group received the questionnaire, and 54 of them responded, sharing their experiences of participating in the PUSHME study “

Since the answers of the questionnaire were anonymized, we cannot examine the response rate by outcome. This is a limitation of the study and is added to the Discussion page 18, line 379-380

“Unfortunately, the questionnaire data was not linked to the individual participant in the study, and therefore it was not possible to check how the response rate varied with outcome.”

Reviewer #2: Borgström Bolmsjö B et al. conducted a trial investigating the impact of health-promoting text messages on patients with hypertension. Their findings indicate that text messages containing lifestyle advice did not significantly affect blood pressure in the general hypertensive population. However, for specific subgroups—namely patients with poor self-rated health (SRH) and those with a sedentary lifestyle—the text messages served as an effective complement to conventional hypertension treatment. These subgroups experienced statistically significant larger mean reductions in diastolic blood pressure (DBP).

Below are some points that should be considered to improve the paper.

2.1. In the Introduction, you mention: "However, despite effective medications, poor adherence to antihypertensive treatment remains a global problem (4)." I suggest also addressing the often-overlooked importance of accurate blood pressure measurement, as this has a significant impact on management strategies (cite PMID: 22902872).

Response:

Thank you for this comment. We have now added this concern to the discussion. Page 16, line 308-309.

“Accurate blood pressure measurement has also been emphasized as being often overlooked in clinical practice, while it could have a considerable impact on management strategies (26).”

2.2 The randomized controlled trial design is appropriate for the research question. The inclusion criteria and the intervention protocol are clearly defined. It would be useful to include more information on how the text messages were created and adapted to individual patients.

Response:

Thank you for this comment. We have now included more information on how the text messages were created and adapted individually. Methods page 6, line 128-135.

“The text messages were developed by the authors based on the Swedish national guidelines for lifestyle habits (9) (10) and edited by experts on lifestyle habits in the regional healthcare administration. Examples of the text messages are shown in Table 1...The text messages were refined after feedback from the participants in the pilot study and were personalized in the matter that each message started with 

---

## [Decision Letter · Decision Letter 1]

19 Nov 2024

Health-promoting text messages to patients with hypertension

- a randomized controlled trial in Swedish primary healthcare

PONE-D-24-17656R1

Dear Dr. Bolmsjo,

We’re pleased to inform you that your manuscript has been judged scientifically suitable for publication and will be formally accepted for publication once it meets all outstanding technical requirements.

Kind regards,

James M Wright

Academic Editor

PLOS ONE

Additional Editor Comments (optional):

Reviewers' comments:

Reviewer's Responses to Questions

**Comments to the Author**

1. If the authors have adequately addressed your comments raised in a previous round of review and you feel that this manuscript is now acceptable for publication, you may indicate that here to bypass the “Comments to the Author” section, enter your conflict of interest statement in the “Confidential to Editor” section, and submit your "Accept" recommendation.

Reviewer #1: (No Response)

Reviewer #2: All comments have been addressed

2. Is the manuscript technically sound, and do the data support the conclusions?

Reviewer #1: (No Response)

Reviewer #2: Yes

3. Has the statistical analysis been performed appropriately and rigorously? 

Reviewer #1: (No Response)

Reviewer #2: Yes

4. Have the authors made all data underlying the findings in their manuscript fully available?

Reviewer #1: (No Response)

Reviewer #2: Yes

5. Is the manuscript presented in an intelligible fashion and written in standard English?

Reviewer #1: (No Response)

Reviewer #2: Yes

6. Review Comments to the Author

Reviewer #1: Thank you for your response to my previous comments.

Regarding Table 4 it is not statistically valid to emphasize only some subgroups - for sex one should present men and women - and the interaction can be tested using the WMD approach as per a meta-analysis without the need for ANCOVA.

Reviewer #2: Thank you to the authors for the revisions made, which I believe have enhanced the quality of the final manuscript.

I have no further comments.

7. PLOS authors have the option to publish the peer review history of their article (what does this mean?). If published, this will include your full peer review and any attached files.

Reviewer #1: No

Reviewer #2: No

---

## [Editor Report · Acceptance letter]

22 Nov 2024

PONE-D-24-17656R1 

PLOS ONE

Dear Dr. Borgström Bolmsjö, 

I'm pleased to inform you that your manuscript has been deemed suitable for publication in PLOS ONE. Congratulations! Your manuscript is now being handed over to our production team.

Kind regards, 

on behalf of

Professor James M Wright 

Academic Editor

PLOS ONE